# Dynamical imprints on precipitation cluster statistics across a hierarchy of high-resolution simulations

Claudia Christine Stephan[1] and Bjorn Stevens[2]

[1]Leibniz Institute of Atmospheric Physics at the University of Rostock, Kühlungsborn, Germany
[2]Max Planck Institute for Meteorology, Hamburg, Germany

**Correspondence:** Claudia Christine Stephan (ccstephan@iap-kborn.de)

**Abstract.** Tropical precipitation cluster area and intensity distributions follow power laws, but the physical processes responsible for this macroscopic behavior remain unknown. We analyze global simulations at ten-kilometer horizontal resolution that are configured to have drastically varying degrees of realism, ranging from global radiative-convective equilibrium to fully realistic atmospheric simulations, to investigate how dynamics influence precipitation statistics. We find the presence of stirring and large-scale vertical overturning, as associated with substantial planetary and synoptic-scale variability, to be key for having cluster statistics approach power laws. The presence of such large-scale dynamics is reflected in steep vertical velocity spectra. Large-scale rising and sinking modulate the column water vapor and temperature field, leading to a heterogeneous distribution of moist and dry patches and regions of strong mass flux, in which large precipitation clusters form. Our findings suggest that power laws in Earth's precipitation cluster statistics stem from the robust power laws of atmospheric motions.

## 1 Introduction

Atmospheric motions span all horizontal scales. Horizontal kinetic energy is associated with geostrophically balanced motions, gravity waves and turbulence, and follows robust power laws when plotted against horizontal wavenumber $\kappa$ (Gage, 1979; VanZandt, 1982; Nastrom and Gage, 1985). The spectrum is shallow at global scales, has a slope of $\kappa^{-3}$ at scales between 10,000–40,000 km, and a slope of $\kappa^{-5/3}$ at shorter scales. The $\kappa^{-3}$ portion of the spectrum is explained by quasi-geostrophic turbulence theory and originates from a downscale enstrophy cascade (Charney, 1971). The mesoscale slope of $\kappa^{-5/3}$ originates from a downscale cascade of wave energy (Cho and Lindborg, 2001; Augier and Lindborg, 2013; Li et al., 2023). Furthermore, in a strongly stratified turbulent flow, one can expect nonlinear interactions between waves and the vortical modes (Müller et al., 1986; Waite and Bartello, 2006; Kitamura and Matsuda, 2010). In contrast to horizontal kinetic energy, the spectrum of vertical kinetic energy is relatively flat (Schumann, 2019), i.e. nearly white. Morfa Avalos and Stephan (2023) showed that the vertical kinetic energy spectrum can be derived from the horizontal kinetic energy spectrum using linear gravity wave theory at large scales and mesoscales, and an incompressible, isotropic scaling of the continuity equation at short scales. While the power laws of atmospheric motions are relatively well understood, this is not the case for moisture fields.

The majority of tropical precipitation clusters follow scale-free frequency distributions for integrated rain rates (Peters et al., 2010; Quinn and Neelin, 2017a) and cluster sizes (Teo et al., 2017). High-resolution simulations (Quinn and Neelin, 2017a)

and Coupled Model Intercomparison Project Phase 5 models (Quinn and Neelin, 2017b) also produce scale free distributions across a wide range of scales. The spectral range over which scale free behavior (or power-law scaling) applies tends to increase with warming. However, the physical origin of the cluster scaling laws remains elusive and motivates our study.

Mathematically, the cluster scaling laws have been described from the perspective of self-organizing criticality (Teo et al., 2017) and percolation theory (Peters and Neelin, 2006; Peters et al., 2009). While these concepts rest on sound mathematical foundations, they do not provide insights on underlying physical mechanisms. Previous studies tried to elucidate the physical processes that matter for the cluster scalings. For instance, Ahmed and Neelin (2019) developed a spatially two-dimensional model (horizontal plane) that represented a number of processes, from the effects of large scale circulation to internal dynamics associated with storm systems. This model combined the weak temperature gradient energy equations with an empirically motivated precipitation parameterization and thus allowed a large variety of sensitivity test. The model reproduced power-law

scaling for realistic model parameter ranges. In our study we use a model that does not rely on empirical representations of the crucial process of moist convection. We focus on exploring the emergence of scaling laws across different planetary configurations. We perform thirteen one-year long global simulations with the Icosahedral Nonhydrostatic model ICON (Hohenegger et al., 2023). The configurations we investigate range from global radiative-convective equilibrium (RCE) simulations to more complex simulations with prescribed sea surface temperatures (SSTs) and land. A relatively coarse horizontal resolution of 10

40  km allows for long integrations, a requirement for sampling the tails in the distributions. We turn convective parameterizations off to explicitly model the main processes responsible for the evolution of the column water vapor (CWV) field.

Focusing on the CWV field is inspired by the study of Li et al. (2022). They performed an aquaplanet simulation and obtained so-called "CWV islands" by cutting the CWV field at the critical moisture value that triggers precipitation. The "island area" is the number of connected pixels exceeding the threshold. The "island volume" is the total CWV above this threshold. They

demonstrated that the areas of these islands agree well with the areas of precipitation clusters, which is of course expected. The same holds for the island volume and area-integrated precipitation, as is also expected. Unexpected was that CWV islands followed scaling laws for a wide range of moisture thresholds: Even when cut at rather dry thresholds where there is no precipitation, the island volumes and areas followed similar scaling laws as if cut at wet thresholds. From a mathematical point of view, this implies that the CWV field must be approximately "self-affine", i.e. upon zooming in or out, it looks the same

after re-scaling the overall amplitude. The height of a self-affine surface is described by $z(\boldsymbol{x}) \sim b^{-h}z(b\boldsymbol{x})$, where $\boldsymbol{x}$ is a location in space, $b$ is a rescaling factor, $h$ is the "roughness exponent" and '$\sim$' denotes statistical equivalence. The roughness exponent $h$ sets the slope of the scaling laws. In other words, it determines how many big clusters should exist for a given number of small clusters or vice versa. Previous work also demonstrated scale free behavior in water vapor variability (Schemann et al., 2013).

As mentioned in the study of Li et al. (2022), the observed $h$ for precipitation clusters is close to the universal prediction of the Kardar-Parisi-Zhang (KPZ) equation. Pelletier (1997) noticed this for statistics of cumulus clouds and both studies suggested that KPZ dynamics could be of relevance. The KPZ equation describes the positive growth of a surface $z(\boldsymbol{x}, t)$ in the

presence of Gaussian white noise $\eta(\boldsymbol{x}, t)$ with a positive mean value. In a frame moving with the surface, the equation reads

$$\frac{\partial z}{\partial t} = \nu \nabla^2 z + \frac{\lambda}{2}(\nabla z)^2 + \eta, \tag{1}$$

where $\nu$ and $\lambda$ are constants. In two spatial dimensions plus time, the solution to the KPZ equation is a surface with $h \approx 0.3867$ (Kardar et al., 1986; Pagnani and Parisi, 2015). Even though the modeling study of Ahmed and Neelin (2019), discussed above, could not isolate a clear physical mechanism to explain the cluster scaling laws, it demonstrated that the observed scalings can be obtained in a spatially two-dimensional model, i.e. without explicitly considering the vertical dimension, which would be consistent with spatially two-dimensional KPZ dynamics. Should the KPZ equation be applicable to Earth's atmosphere, then Earth's motion spectrum would have to match the required structure of the noise term $\eta$. While a physical interpretation of the $\eta$ term is not the goal of our study, we want to test if changes in the atmospheric motion spectrum go along with changes in the statistics of precipitation clusters. If this were systematically the case, then it would be strong evidence that the characteristics of precipitation clusters result from the robust scaling laws of atmospheric horizontal (or vertical) kinetic energy.

In this study, we therefore consider atmospheres that differ substantially in their dynamic characteristics. The planetary configuration determines the scaling behavior of the atmospheric motion field. Our main result is that self-affinity in the CWV field and scaling laws in precipitation clusters only emerge in atmospheres with large-scale mixing, which suggests that precipitation clusters do indeed inherit their scalings from atmospheric motion spectra.

Section 2 describes our data and methods, Section 3 presents the results, and Section 4 contains a brief conclusion.

## 2 Data and methods

### 2.1 Numerical simulations

All simulations have a 10 km horizontal grid spacing and 75 terrain-following height levels with a model top at 48 km. A Rayleigh damping layer starts at 19 km. Microphysics, radiation and turbulence use the default Sapphire physics schemes (Hohenegger et al., 2023). Although a horizontal resolution of 10 km may be generally thought to be too coarse to explicitly represent convection, a number of studies show that it better and more physically represents convective processes than existing parameterizations (Holloway et al., 2012; Vergara-Temprado et al., 2020; Takasuka et al., 2024). Bravo et al. (2024) tested the convergence of aquaplanet simulations using ICON with the same physics package as our study. They refined the horizontal grid from 160 km down to 1.25 km and report that tropical precipitation and precipitable water converge already at 10 km with only small changes towards finer resolution.

**CTL** is the control simulation, which is initialized on January 1, 1979 from IFS operational analysis and has daily varying prescribed SSTs interpolated from the monthly SST and sea ice concentration boundary conditions for AMIP II simulations (Taylor et al., 2000). Globally and annually averaged concentrations of greenhouse gases (Meinshausen et al., 2016) are prescribed from values taken from the respective year. Ozone is included with year 2014 values and varies spatially on a grid with $2.5°$ resolution in longitude and $1.675°$ resolution in latitude and on a monthly timescale, based

on input4MIPs data (input datasets for Model Intercomparison Projects). Aerosols are specified from a climatological data set (Kinne, 2019), which provides monthly data on a 1° grid. External physical parameters for the land surface are based on Hagemann and Stacke (2015). The CTL simulation is a member of one of three classes of simulations. The remaining twelve simulations differ as follows.

$CTL_{zm}$ is like CTL except that it uses constant and zonally averaged SSTs, which correspond to the time-averaged zonal-mean SSTs of CTL, i.e. SSTs of 1979. Land is like in CTL. CTL and $CTL_{zm}$ are the only simulations with a diurnal and annual cycle. By comparing to CTL we can assess the importance of SST variability for shaping the water vapor and precipitation fields.

**RCE** is a radiative-convective equilibrium (RCE) simulation, i.e. non-rotating with constant insolation. It follows the 300 K RCEMIP protocol (Wing et al., 2018). RCE simulations have been employed for decades as a standard idealized setup to study processes in the tropical atmosphere (e.g., Bretherton et al. (2005)).

$RCE_s$ is like RCE but has the meridionally varying 'Qobs' SST profile (Neale and Hoskins, 2000) with a maximum SST of 27°C at the equator that decreases to a minimum SST of 0°C which is held fixed poleward of 60° in each hemisphere. This simulation lets us test the impact of a more realistic meridional SST profile on the distribution of moisture.

$RCE_r$ is like RCE but rotates at the angular velocity of Earth. This type of simulation is described in detail in Shi and Bretherton (2014). We conduct this simulation to test the effect of a varying Coriolis parameter, which would, among other phenomena, allow the generation of cyclones in the extratropics.

$RCE_f$ is like RCE but rotating with the Coriolis parameter set globally to its value at 45°N. A simulation of this type, albeit with a Coriolis parameter for 10°N, is described in Reed and Chavas (2015). In such a simulation cyclones form everywhere. We thus expect a distribution of moisture and precipitation that is quite different from reality. We here ask if some properties of precipitation clusters still persist in this extreme setup.

**AP** is an aquaplanet (AP) simulation that combines $RCE_r$ and $RCE_s$, i.e. it rotates like Earth and the SST is prescribed following $RCE_s$. This experiment is relatively close to $CTL_{zm}$ except it lags land and has a constant insolation. By comparing with $CTL_{zm}$ we can test the importance of the additional heterogeneity in $CTL_{zm}$. Since the AP setup is the idealized experiment closest to Earth, we perform six additional AP simulations.

$AP_{t+}$ **and** $AP_{t-}$ are like AP but with +5 or -5 K added to the SST everywhere including the poles. We here test how precipitation cluster statistics depend on temperature. Based on Quinn and Neelin (2017b) we would expect the spectral range over which power-law scaling applies to increase with warming.

$AP_{g+}$ **and** $AP_{g-}$ have gravity increased or decreased, respectively, by 20% relative to 9.81 m s$^{-2}$. This changes the dry adiabatic lapse rate $\mathrm{d}T/\mathrm{d}z = -g/c_p$, with $T$ temperature, $z$ height and $c_p$ the isobaric specific heat capacity. $AP_{g-}$ will thus be warmer. The $AP_{g+}$ and $AP_{g-}$ experiments are an alternative way of changing the temperature and moisture content of the atmosphere without changing the SST.

$\mathbf{AP_{fg+}}$ and $\mathbf{AP_{fg-}}$ are like $AP_{g+}$ and $AP_{g-}$ except that the square of Earth's angular velocity is also changed by plus or minus 20%, respectively, so that the centrifugal force partly compensates the increase in gravity.

The model output consists of instantaneous fields, written at daily intervals (00 UTC), of precipitation, temperature, CWV and vertical velocity at 5 km. The first 100 days of the simulations are discarded for spin-up. After 100 days, global mean outgoing longwave radiation, CWV and precipitation fluctuate around their equilibrium. The last 260 days are subject to the analysis. Although the characteristics of the precipitation clusters are stable during these 260 days, we need this amount of data to sample the tails of the distributions.

## 2.2 Observations

We analyze instantaneous CWV at hourly intervals from the ERA5 reanalysis (C3S, 2017), and reprocessed and bias-corrected 30-min accumulations of precipitation at hourly intervals from CMORPH (Xie et al., 2019) from 1998 to 2021. We use hourly data for the observations, as unlike in the majority of our numerical simulations, there is a diurnal cycle in reality. ERA5 is provided on a $0.28125° \times 0.28125°$ latitude-longitude grid with a resolution of 31 km at the equator. CMORPH is provided on a $0.0727° \times 0.0727°$ latitude-longitude grid which corresponds to a resolution of 8 km at the equator. Thirty minutes is the shortest available accumulation time for CMORPH. For the model output we chose to analyze instantaneous fields to facilitate the interpretability when connecting different variables, since each variable has a different de-correlation time scale. For simplicity and to better emphasize the difference with the modeling results, we refer to ERA5 as observations.

## 2.3 Cluster and island properties

This study focuses on the tropical regions and so only output between 25°S and 25°N is analyzed, and for this purpose regridded from the native 10 km grid to a $0.1° \times 0.1°$ latitude-longitude grid. We process ERA5 and CMORPH in the same way.

We define grid points as precipitating when precipitation rates exceed a threshold of 2 mm h$^{-1}$. Li et al. (2022) used 0.7 mm h$^{-1}$, but report that their results were not sensitive to varying the threshold between 0.1 and 2.5 mm h$^{-1}$. Their study also differs in that they considered 3-hourly averages. Models with an explicit representation of moist convection at what is still rather coarse resolution tend to heavy rain (Becker et al., 2021). We thus select 2 mm h$^{-1}$ closer to the upper bound reported by Li et al. (2022). As we show later, this threshold also produces the expected scalings in observations. A cluster is then defined as all pixels that are connected by at least one common edge, i.e. touching corners do not count. We give the area, $A$, in units of pixels, where one pixel measures $0.1° \times 0.1°$. Latitudinal distortions are small due to confining the analysis to the tropics and hence ignored. For the perimeter length $\lambda$ we report the number of pixel edges, such that a single isolated precipitating pixel would have a perimeter of 4. Precipitation cluster volume $I$ is defined as instantaneous precipitation rates integrated over a cluster, i.e. total cluster precipitation, and thus has units of mm h$^{-1}$ pixel. We denote the exponent of the area frequency distribution by $\alpha$, such that the probability to encounter a cluster with area $A$ is given by $P(A) \sim A^{\alpha}$. Similarly, for the volume $I$, $P(I) \sim I^{\beta}$. Perimeter $\lambda$ and volume $I$ are related to area through the fractal dimensions $\delta_{\lambda}$ and $\delta_{\mu}$: $\lambda = A^{\delta_{\lambda}/2}$ and $I = A^{\delta_{\mu}/2}$.

We also analyze areas ($A$) and volumes ($V$) of CWV islands. Note that we use the symbol $I$ for the volume of precipitation clusters and $V$ for the volume of CWV islands, as these quantities have different units. The volume of a CWV island is the total CWV above a threshold inside the corresponding island area, i.e. for each CWV island we sum the CWV of all pixels above the threshold after subtracting the threshold value. We cut the CWV field at three different thresholds: the 40th and 80th percentiles of the tropical CWV frequency distribution of each data set, and the critical threshold, defined as the CWV where precipitation rates reach 2 mm h$^{-1}$ on average. These critical threshold values are discussed below in Section 3.2. We confine our analysis to columns over the ocean to increase the chance of finding universal behavior across the domains, many of which do not have land, but this only influences the analysis of the CTL simulations and the observations. Areas of CWV clusters are defined in the same way as for precipitation.

To estimate the scaling exponents $\alpha$ and $\beta$ and the fractal dimensions $\delta_\lambda$ and $\delta_\mu$, we bin the area and volume data into logarithmic bins and perform a linear regression in log-log space. The logarithmic binning reduces noise in the tails of the distributions. The start and end values of the regression range are chosen to maximize the linear correlation coefficient between the data and the fit, but for a valid fit we require the linear correlation coefficient to be at least 0.999. Additionally, the fit must use at least a quarter of the logarithmic bins. If these criteria cannot be fulfilled, no spectral slope is determined. In the corresponding plots we highlight data points when they fall within 10% of the best-fit line.

## 3 Results

### 3.1 Dynamics on different scales

Figure 1 shows the global CWV field on day 100 of the simulations and on a randomly chosen date in ERA5 (denoted as Obs.). It documents substantial differences in planetary and synoptic scale variability and in the associated stirring and vertical overturning (macroturbulence) of the atmosphere. In the following we will first discuss the patterns in the CWV fields and then quantify how the simulations differ in their tropical vertical motions.

The control simulation CTL with realistic SSTs, land and a diurnal cycle, shows some characteristics that are also seen in observations, such as enhanced moisture above the warm SSTs of the western tropical Pacific. As expected, this local maximum is absent when zonal mean SSTs are prescribed (CTL$_{\mathrm{zm}}$). The aquaplanet simulation AP has less variability compared to CTL and CTL$_{\mathrm{zm}}$. Still, the effects of planetary and synoptic scale disturbances on the CWV field of AP visibly define a tropical margin.

Moist and dry patches form globally in the radiative-convective equilibrium simulation RCE, indicative of convective self-aggregation (Muller and Held, 2012). RCE$_{\mathrm{s}}$, which has a meridional SST gradient, produces a band of high CWV at the equator where SSTs are maximum. The CWV band in RCE$_{\mathrm{s}}$ has structures on finer scales compared to RCE. RCE$_{\mathrm{r}}$, which rotates like Earth, and RCE$_{\mathrm{f}}$, where the Coriolis parameter is globally set to its value at 45°N, produce tropical storms where the Coriolis force is nonzero (Held and Zhao, 2008). In RCE$_{\mathrm{r}}$ the CWV field is relatively smooth in the tropics, where the Coriolis parameter is not large enough to support the formation of tropical storms. The CWV field of RCE$_{\mathrm{r}}$ shows planetary and (to a lesser extent) synoptic scale variability in the zonal direction. This is consistent with the study of Arnold and Randall

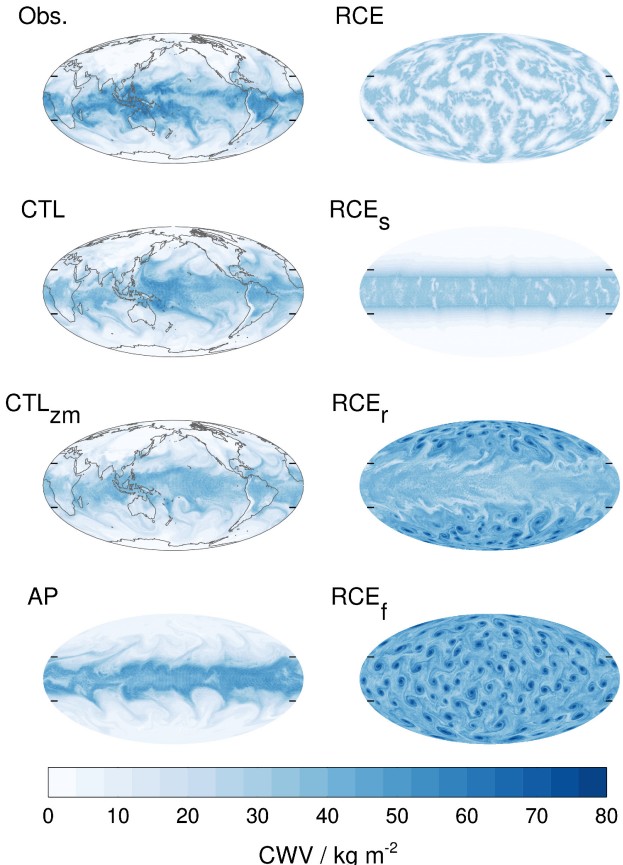

**Figure 1.** Global maps of CWV. Data from ERA5 (Obs.) show March 1, 2011 00 UTC and the seven experiments show day 100 of the simulations (after spinup). Short lines at the right and left edge of each map mark the tropics ($\pm 25°$).

(2015), who contrasted tropical variability in a rotating setup similar to our $RCE_r$ simulation with a non-rotating simulation. Chavas and Reed (2019) examined the sensitivity of such simulations to rotation rate and Earth's radius. We do not show the extra AP simulations in Fig. 1, because their structures look very similar to AP. Differences between the AP experiments are discussed in Appendix A.

Recall that we aim at establishing a link between the statistical properties of the CWV field and those of atmospheric motions. For this reason we compute how vertical kinetic energy is distributed as a function of the zonal wavenumber $\kappa$. A steep slope (large negative values) indicates that motions on large horizontal scales (small $\kappa$) are more energetic than motions on short horizontal scales (large $\kappa$). When the slope is shallow (small negative values), then motions on short scales become relatively more energetic. Hence, vertical velocity spectra are useful for comparing the prevalence of different scales of vertical motion

between the simulations. The spectrum of CTL is shown in Fig. 2. The spectra of the other simulations look similar, differing only in the spectral slope (Appendix B). The slopes of the simulated vertical velocity spectra at zonal wavenumbers $8 < k < 300$

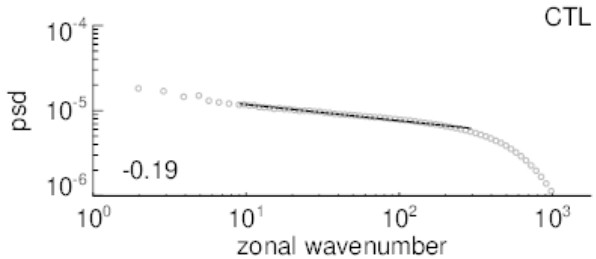

**Figure 2.** Meridionally averaged power spectral density in units of $m^2 \ s^{-2}$ of vertical velocity at 5 km height for CTL. The slope shown inside the panel is computed for zonal wavenumbers $8 < k < 300$, shown in black.

vary between -0.05 (RCE and $RCE_s$) and -0.24 ($RCE_f$). Thus, coherent large-scale overturning is most ubiquitous – relative to other scales – in $RCE_f$, followed by the CTL simulations (CTL: -0.19, $CTL_{zm}$: -0.17), followed by the AP simulations (between -0.09 and -0.16). The remaining RCE experiments have flatter slopes, which is consistent with a preference for overturning on short horizontal scales and relatively weak large-scale dynamics. Next, we examine what these differences imply for the scaling of precipitation clusters.

### 3.2 Mass flux and cluster size

Figure 3 shows a snapshot of a random scene taken from the different configurations. In panel (a) we can clearly see the difference between, for instance, RCE and $RCE_f$, the two simulations that differ most in the slope of vertical kinetic energy. RCE contains either dry regions without precipitation or moist regions with isolated convection. Here, vertical velocity has most energy on short scales. $RCE_f$, on the other hand, forms tropical storms and generates strong vertical motions on large scales. In general, in the simulations with less pronounced large-scale dynamics, isolated, small-scale convection prevails. Panel (b) shows the product of CWV and vertical velocity at 5 km height for the same time as in Fig. 3a. As expected, the product correlates well with the CWV field itself, i.e. ascent occurs predominantly in moist regions. Large positive values in Fig. 3b are indicative of strong mass flux. Stronger mass flux can be realized either through stronger ascent within same-sized convective areas, or through an increase in area coverage. Previous studies demonstrated that nature follows the second option due to microphysical constraints (Doneaud et al., 1984; Nuijens et al., 2009; Parodi et al., 2011; Fildier et al., 2017). We will now investigate if this is also the case in our simulations.

Because warmer air can hold more moisture, the CWV threshold for the onset of precipitation depends strongly on tropospheric bulk temperature ($T_b$), which is defined as mass weighted temperature between 1 and 10 km height. In each simulation, $T_b$ varies in space and time, but due to the different setups of the experiments, the average $T_b$ is also different from experiment to experiment. The most frequently occurring value of $T_b$, the mode $T_m$, is shown on the y-axis of Fig. 4. The corresponding x-axis shows the critical CWV, defined as the CWV where precipitation rates reach on average 2 mm $h^{-1}$ at points with the respective $T_m$. Since we are not interested in the values of $T_b$ and CWV themselves, but want to focus on the structure of the

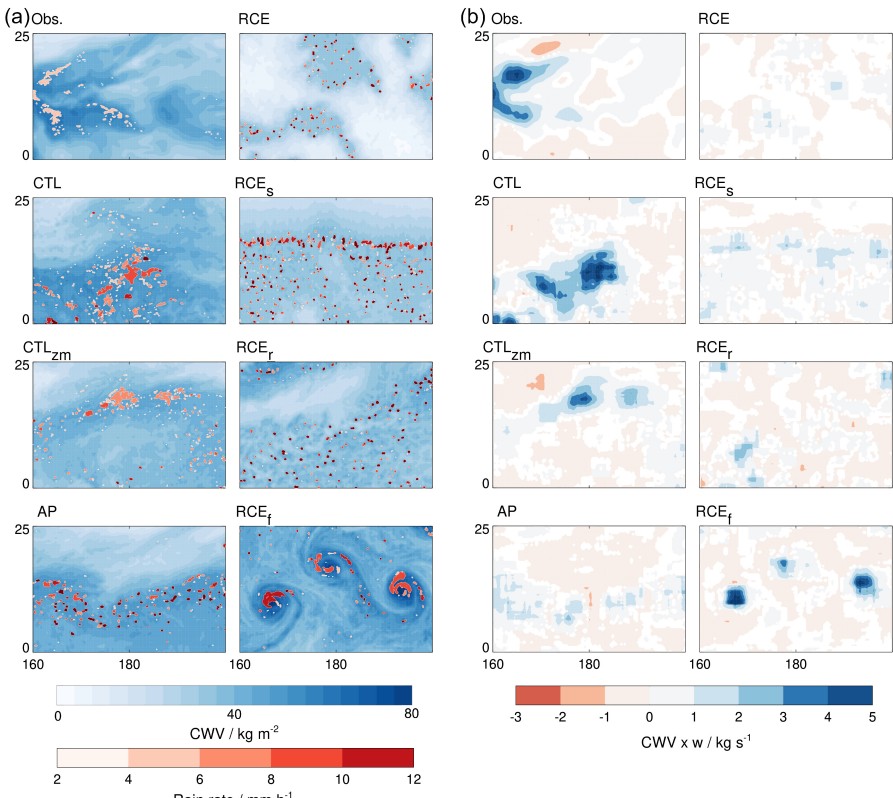

**Figure 3.** Snapshots of moisture fields. (a) shows CWV and cluster-mean precipitation rate on day 100 of the simulations and on March 1, 2011 00 UTC in observations. The y-axis shows latitude in °N and the x-axis shows longitude in °E. (b) shows for the same times CWV multiplied by the vertical velocity at 5 km height. Before multiplication both fields were smoothed with a 390 km boxcar filter. Values between -0.1 and 0.1 kg s$^{-1}$ are colored white.

CWV field and its relationship to dynamics, we rescale $T_{\mathrm{b}}$ and CWV to achieve a better comparability of the simulations. We rescale temperature by subtracting $T_{\mathrm{m}}$ from $T_{\mathrm{b}}$ and we define reduced CWV as $(\mathrm{CWV}-\mathrm{CWV}_{\mathrm{crit}})/\mathrm{CWV}_{\mathrm{crit}}$, following Peters and Neelin (2006), who showed that precipitation onset curves collapse when plotted against reduced CWV. This is also true for our simulations. Figure 5 shows the precipitation onset lines in the original phase space (CWV and $T_{\mathrm{b}}$) and rescaled phase space (reduced CWV and $T_{\mathrm{rel}} = T_{\mathrm{b}} - T_{\mathrm{m}}$). Except for dry or moist outliers, the onset curves collapse onto the same line in the normalized phase space. This is particularly clear for the AP simulations (Fig. 5c,d). For the other simulations (Fig. 5a,b) the change is less impressive because these simulations differ widely in their temperature distributions (the range of simulated temperatures is shown as vertical lines next to the y-axis). Given that $\mathrm{CWV}_{\mathrm{crit}}$ scales approximately linearly with $T_{\mathrm{m}}$ (Fig. 4), it may seem surprising that the onset curves collapse only in normalized phase space. An explanation can be found in Fig. C1. For example, from $\mathrm{AP}_{\mathrm{t}-}$ to $\mathrm{AP}_{\mathrm{t}+}$ the temperature probability density distribution shifts to higher values with little change to

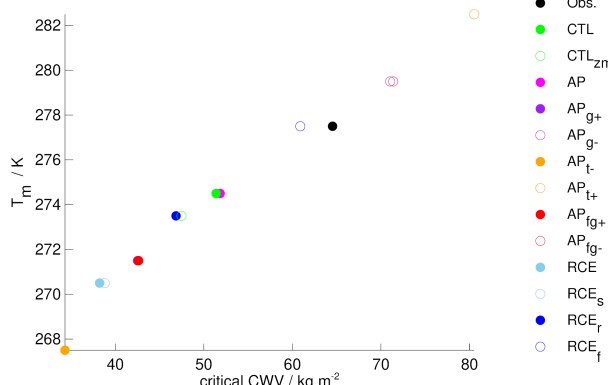

**Figure 4.** For each data set symbols mark $CWV_{crit}$ corresponding to the most frequently occurring bulk temperature bin $T_m$.

its shape. In contrast, the CWV distribution does not merely shift to higher values, but also broadens. This results in a flatter slope of the onset curve. Rescaling the CWV accounts for the width of the CWV distribution.

Fig. 6 shows the dependence of cluster area on the maximum reduced CWV inside a cluster for our data sets (black) and in addition it shows the average precipitation rate inside clusters (blue), i.e. volume divided by area. With increasing $CWV_{crit}$, area size increases more rapidly than the average precipitation rate. However, the difference in growth rates is much more pronounced in CTL and $RCE_f$, whereas it is substantially smaller in the other RCE simulations. This is again due to the presence of stronger large-scale dynamics and greater mass flux.

We now turn to the question to what extent different dynamics influence the area-size distributions of precipitation clusters and CWV islands.

### 3.3  Cluster spectra

Figure 7 shows the occurrence frequency spectra of areas for CWV islands at different thresholds and precipitation clusters. The x-axis is logarithmic, as is the y-axis. The latter is not shown, because we are only interested in the spectral slope. The corresponding slopes of all data sets are summarized in Fig. 8a whenever a fit was possible. The occurrence frequency spectra for volumes are shown in Appendix D.

Focusing first on observations, CMORPH precipitation follows a slope of -1.79 for area (gray star in the leftmost column of Fig. 8a). The CWV islands of ERA5 cannot be expected to follow any scaling law at fine scales, as ERA5's horizontal resolution is a factor of ∼4 coarser than that of CMORPH. For the 40% and 80% thresholds, the area scalings in ERA5 are flatter (40%: -1.62, 80%: -1.41) than those of precipitation, but at the critical threshold the slope is steeper (-1.87). Differences in the slopes of CWV and precipitation volumes (Fig. 8b) are similar to those in the area distributions.

We now turn to the simulations, focusing first on the control simulations. By visual inspection of Fig. 7, CTL and $CTL_{zm}$ show the best overall agreement of CWV and precipitation spectra with observations, when we consider all scales. It should be

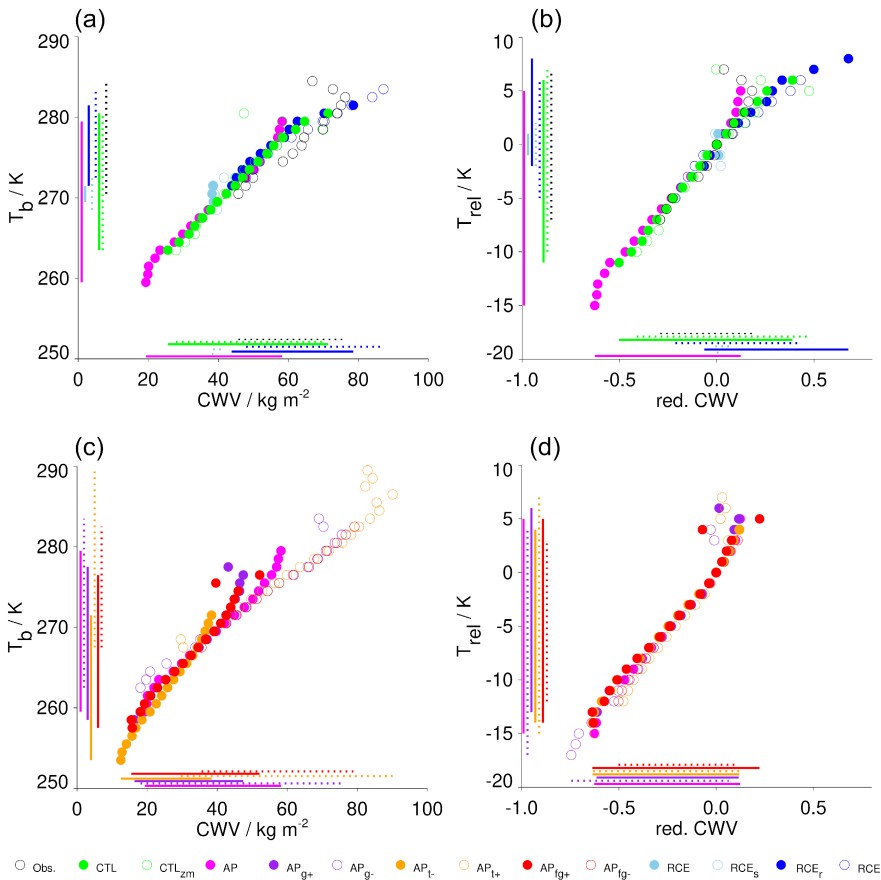

**Figure 5.** Precipitation onset curves. In (a) and (c) an open or filled circle marks the average CWV at which precipitation rates reach 2 mm h$^{-1}$ for each bulk temperature $T_b$ (y-axis). Lines along the axes mark the range covered by the respective 2 mm h$^{-1}$ isoline of a simulation. (b) and (d) repeat (a) and (c) in the phase space of reduced CWV and relative temperature $T_{rel} = T_b - T_m$.

noted that the cluster statistics for CTL and CTL$_{zm}$ suffer from the fact that clusters touching land have to be discarded. This causes noisy tails. In ERA5 this effect is compensated by the amount of data spanning 24 years. It appears that the simulated spectra, particularly those of precipitation, have a slight bias towards small scales. A local maximum in the spectra can be seen even in CTL. It is common that simulations tend to preferentially form small precipitation clusters that rain too heavily when using an explicit representation of moist convection at what is still rather coarse resolution (Becker et al., 2021). Fig. 3a also illustrates this point. It is conceivable that looking at instantaneous data of the simulations versus 30-min averaged data of CMORPH explains some of these biases. Moreover, precipitation is not directly measured by remote sensing instruments but estimated from infrared measurements of column condensate path, which might introduce some additional smoothing (Pradhan et al., 2022).

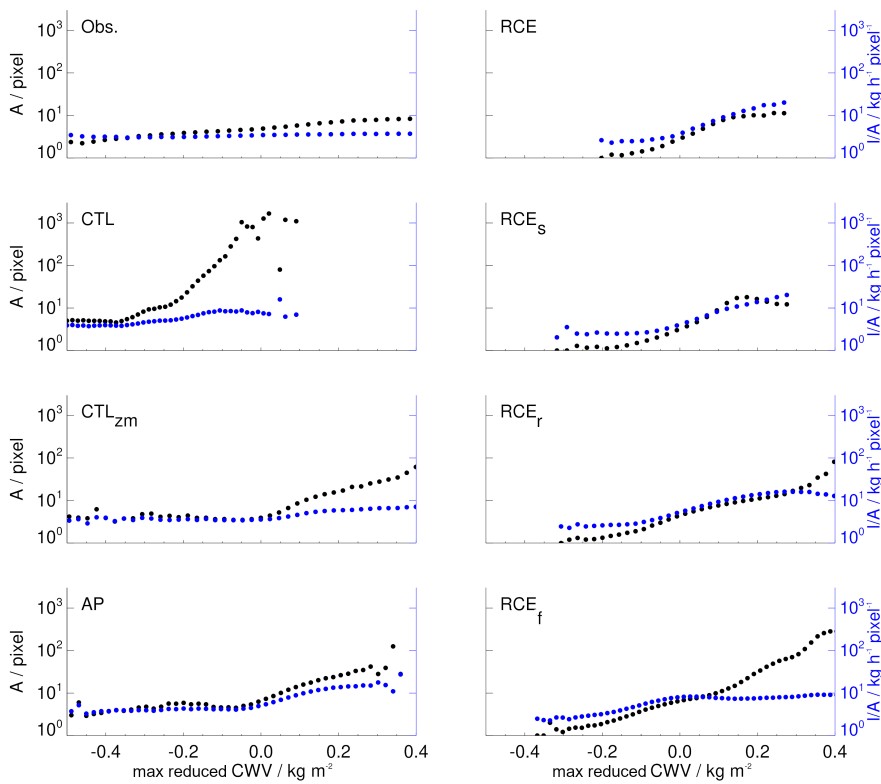

**Figure 6.** The average precipitation cluster area (black) and volume divided by area (blue) as function of the maximum reduced CWV value inside a cluster.

The curves for the CWV areas of all AP simulations resemble the CTL simulations, with precipitation areas following steeper curves than CWV island areas, as in CTL.

Finally, the RCE simulations vary greatly in the degree of similarity between spectra at different CWV thresholds and those of precipitation.

We now turn to question if changes in the atmospheric motion spectrum – in this case diagnosed as the vertical velocity spectrum becoming shallower or steeper – change the statistics of CWV islands and precipitation clusters. This implicitly includes the question whether or not precipitation cluster statistics inherit their properties from the statistics of the CWV distribution, as the study by Li et al. (2022) proposed. From the above discussion of Fig. 7 we see that curves of CWV at different thresholds of precipitation align better in some experiments than in others. This says that precipitation does not inherit its statistics from universal CWV statistics in all experiments. However, in some experiments the 4 colored curves in Fig. 7 have similar shapes, which points to self-affinity of the CWV field.

It is straightforward to understand why the curves diverge in some experiments. In RCE, RCE$_s$ and RCE$_r$, precipitation occurs in moist patches as isolated, small scale convection, creating updrafts on small horizontal scales. These motions do

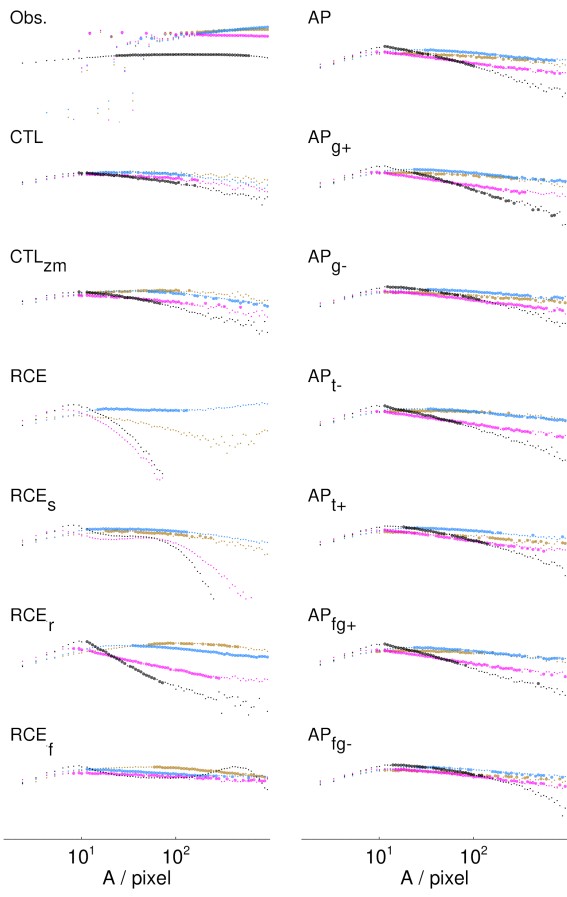

**Figure 7.** Occurrence frequencies of areas belonging to precipitation clusters (black) and CWV islands at the 40% threshold (brown), the 80% threshold (blue), and the critical threshold (magenta). All curves are compensated by $A^{1.79}$. Thick dots mark the intervals with the best power law (Section 2.3).

not require large-scale overturning but can be compensated by downwelling on similarly small scales. Hence, in RCE, $RCE_s$ and $RCE_r$, the small-scale dynamics decouple from the large-scale distribution of the CWV field. Weak and slowly varying

overturning only shapes large-scale dry and moist patches. In these experiments there is no self-affinity of the CWV distribution.

Our results show that a better match between the scalings of CWV islands at all thresholds and precipitation go along with steeper vertical velocity slopes. This also holds for the AP experiments. The warmer AP simulations have a better match and steeper slopes of vertical kinetic energy ($AP_{g-}$: -0.15, $AP_{t+}$: -0.12, $AP_{fg-}$: -0.16) than AP (-0.11) and than the colder experiments, which have flatter slopes ($AP_{g+}$: -0.09, $AP_{t-}$: -0.10, $AP_{fg+}$: -0.09).

Lastly, we now briefly turn to the roughness exponent $h$. The KPZ equation (Eq. 1) predicts $h \approx 0.3867$. For self-affine surfaces, the slopes for area and volume occurrence ($\alpha$, $\beta$) and the fractal dimensions ($\delta_\lambda$, $\delta_\mu$) are related to $h$. The area scaling exponent $\alpha$ is given by $\alpha = h/2 - 2$ (Pelletier, 1997). Li et al. (2022) derived for $\beta$ the relationship $\beta = -4/(2 + h)$, Kondev

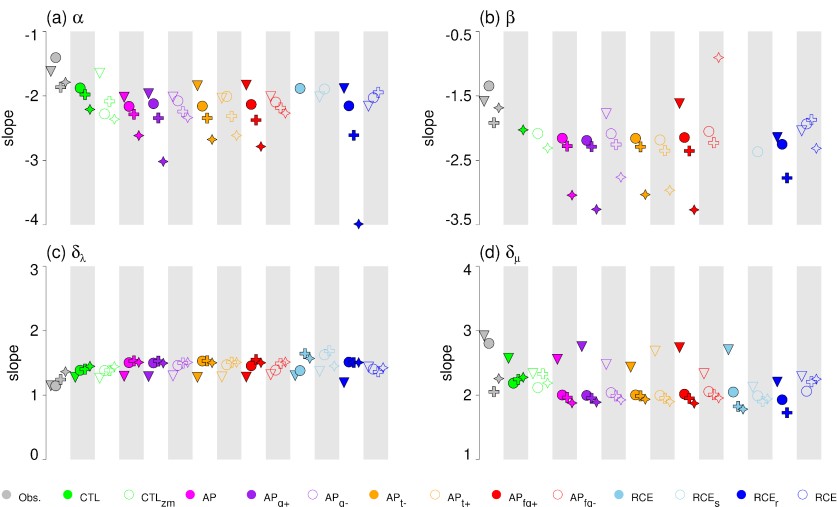

**Figure 8.** Slopes obtained from fitting a scaling law to the data shown in Figs 7 (panel a), D1 (panel b), D2 (panel c) and D3 (panel d). Each column shows a different data set identified by filled or unfilled colors as shown by the legend at the bottom. Each column can have four symbols, where the star stands for precipitation, the downward pointing triangle for the 40% CWV threshold, the circle for the 80% threshold, and the plus for the critical threshold. A missing symbol implies that a fit could not be obtained.

and Henley (1995) showed $\delta_\nu = (3 - h)/2$, and Li et al. (2022) $\delta_\mu = 2 + h$. Table 1 lists $\alpha$, $\beta$, $\delta_\lambda$ and $\delta_\mu$ for different values of the roughness exponent $h$. Plots of the fractal dimensions are shown in Appendix D.

**Table 1.** Scaling exponents and fractal dimensions. Rows list the scaling exponents ($\alpha$, $\beta$) and fractal dimensions ($\delta_\lambda$, $\delta_\mu$) for different roughness exponents ($h$), and those obtained for CMORPH and the CTL simulation.

|  | $h = 0.4$ | $h = 0.3$ | $h = 0$ | $h = 1$ | CMORPH | CTL |
|---|---|---|---|---|---|---|
| $\alpha$ | -1.80 | -1.85 | -2.00 | -1.50 | -1.79 | -2.21 |
| $\beta$ | -1.67 | -1.74 | -2.00 | -1.33 | -1.69 | -2.03 |
| $\delta_\lambda$ | 1.30 | 1.35 | 1.50 | 1.00 | 1.36 | 1.45 |
| $\delta_\mu$ | 2.40 | 2.30 | 2.00 | 3.00 | 2.26 | 2.28 |

The area and volume scaling exponents ($\alpha$, $\beta$) of CMORPH precipitation clusters are consistent with a roughness exponent of about 0.4. The fractal dimensions for CMORPH are more consistent with $h \simeq 0.3$. As we can already see from Fig. 8, CTL deviates from these values. In general, Fig. 8 shows that observed and simulated precipitation clusters (star symbols) agree much better in terms of their fractal dimensions than they do in terms of area or volume distributions (see also Figs D2 and D3). The better agreement in $\delta_\lambda$ and $\delta_\mu$ compared to $\alpha$ and $\beta$ between the simulations suggests that the fractal dimensions might be controlled by turbulence, as was also proposed by previous studies (Garrett et al., 2018; Siebesma and Jonker, 2000).

## 4 Conclusions

Based on simulations of Earth-like and un-Earth-like planets we searched for ingredients that are required to have precipitation clusters follow scaling laws. Key is the presence of stirring and large-scale vertical overturning as associated with substantial planetary and synoptic-scale variability. The presence of such large-scale dynamics is reflected in steep vertical velocity spectra. Large-scale rising and sinking modulate the column water vapor (CWV) and temperature field, leading to a heterogeneous distribution of moist and dry patches and regions of strong mass flux, in which large precipitation clusters form. A dearth of large-scale structures, as reflected in flat vertical velocity spectra, is associated with a decoupling of the large-scale distribution of CWV and precipitation. In this latter scenario, weak overturning shapes large-scale dry and moist patches, and only weak mass flux is present in the moist patches. Precipitation is then realized as unorganized isolated convection.

The global motion spectrum is strongly shaped by the distribution of orography and the presence of rotation and differential heating, with convection contributing mainly to the mesoscale motion spectrum (Stephan et al., 2019a, b; Köhler et al., 2023). Horizontal motion spectra follow robust scaling laws (Stephan et al., 2022), which also determine the spectra of vertical motion (Morfa Avalos and Stephan, 2023) and horizontal divergence (Stephan and Mariaccia, 2021). Our finding that the structure of Earth's motion field is important to having precipitation clusters follow scaling laws implies that any potential applicability of the Kardar-Parisi-Zhang (KPZ) equation to the problem of Earth's CWV field relies on the coincidence of the noise term $\eta$ in Eq. 1 agreeing with the properties of Earth's motion spectra.

Our study shows that precipitation cluster statistics derive their apparent universality from the robust spectral characteristics of atmospheric wave and turbulence dynamics. This is plausible because there has to be a global organization mechanism for precipitation clusters if they obey scaling laws. While this result is new to the best of our knowledge, it is only a qualitative statement and future studies are needed to address the causal chain of underlying mechanisms. Interesting open questions that our study inspires are, for instance:

- What is the required slope in vertical velocity to achieve self-similarity in the CWV distribution and what would happen if the slope were steeper?

- Is there an important feedback from convection, i.e. is the generation of waves and vertical motions by condensation relevant for the relationship between atmospheric motions and the distribution of CWV?

- Why is the roughness exponent close to the prediction of KPZ dynamics? Is it a coincidence or can we interpret the $\eta$ term of the KPZ equation in a physically meaningful way?

We are confident that progress is possible with well-designed numerical experiments. A starting point could be to prescribe atmospheric motions and use water vapor as a passive tracer.

*Code and data availability.* CMORPH data can be obtained from the National Centers for Environmental Information's National Oceanic and Atmospheric Administration. The ERA5 reanalysis is produced and made publicly available by the ECMWF. The ICON model (ICON-

partnership, 2024) is open source. More information about ICON is available at https://www.icon-model.org/. The versions of the code used for the simulations analyzed here are uniquely identified by their git hashes. They are: 0a47ce01ee3c6dada56e21b17da66f54499db3e3 for CTL and CTL$_{zm}$ and 0d09a39ed159c51cf6f8ce5c654cf206c8c037c5 for all other simulations.

## Appendix A: Aquaplanet simulations

In the extra aquaplanet simulations the SSTs were globally increased ($AP_{t+}$) or decreased ($AP_{t-}$), gravity was increased ($AP_{g+}$) or decreased ($AP_{g-}$), and gravity together with the Coriolis parameter were increased ($AP_{fg+}$) or decreased ($AP_{fg-}$). The zonal mean CWV of the AP simulations is compared in Fig. A1. $AP_{t+}$ is warmer than AP and therefore contains more CWV in the tropics. $AP_{t-}$ behaves in the opposite way. In addition, Fig. A1 shows that the CWV peaks are further apart in $AP_{t+}$, indicative of broader tropics, and less apart in $AP_{t-}$. $AP_{g-}$ and $AP_{fg-}$ look nearly the same. They contain almost as much CWV as $AP_{t+}$, with CWV maxima closer to the equator than AP. Similarly, $AP_{g+}$ and $AP_{fg+}$ are colder than AP with peaks further away from the equator.

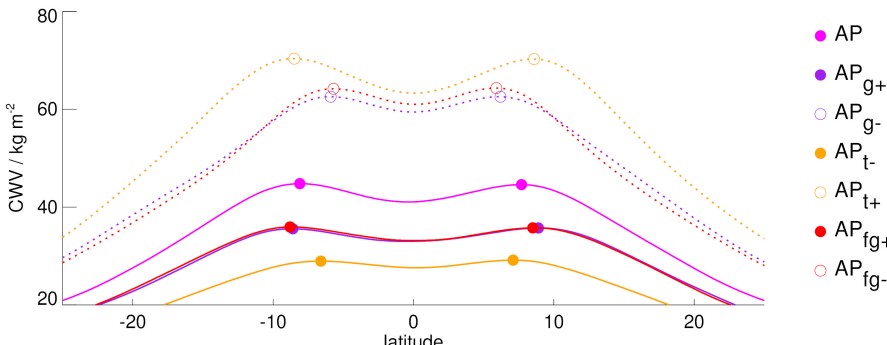

**Figure A1.** Zonal-mean time-mean CWV in the tropics as simulated by the aquaplanet simulations. Open or filled circles mark the respective maximum in each hemisphere to facilitate the comparison.

# Appendix B: Vertical velocity spectra

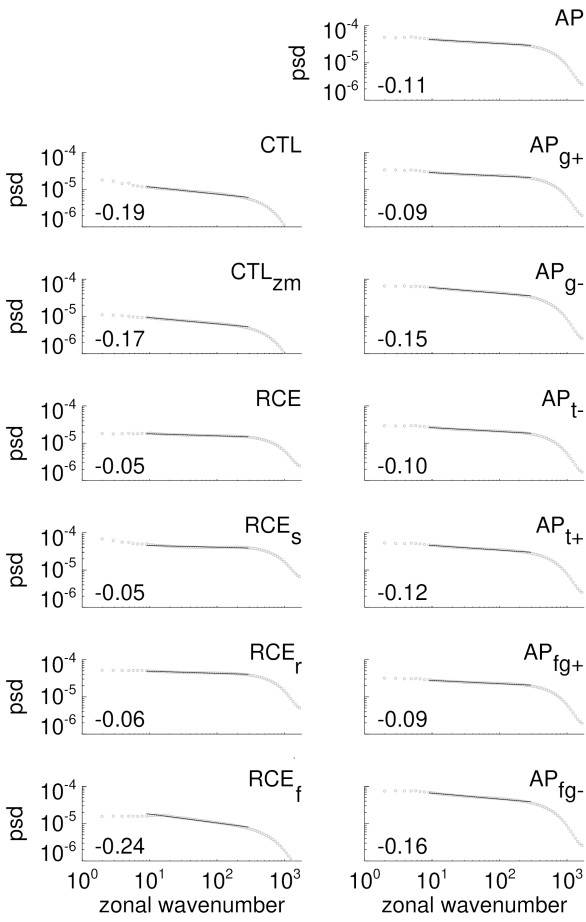

**Figure B1.** Meridionally averaged power spectral density in units of m$^2$ s$^{-2}$ of vertical velocity at 5 km height. Slopes shown inside the panels are computed for zonal wavenumbers $8 < k < 300$, shown in black. All displayed decimal places are one order of magnitude larger than their one-sigma uncertainty estimates.

## Appendix C: The phase space of temperature and CWV

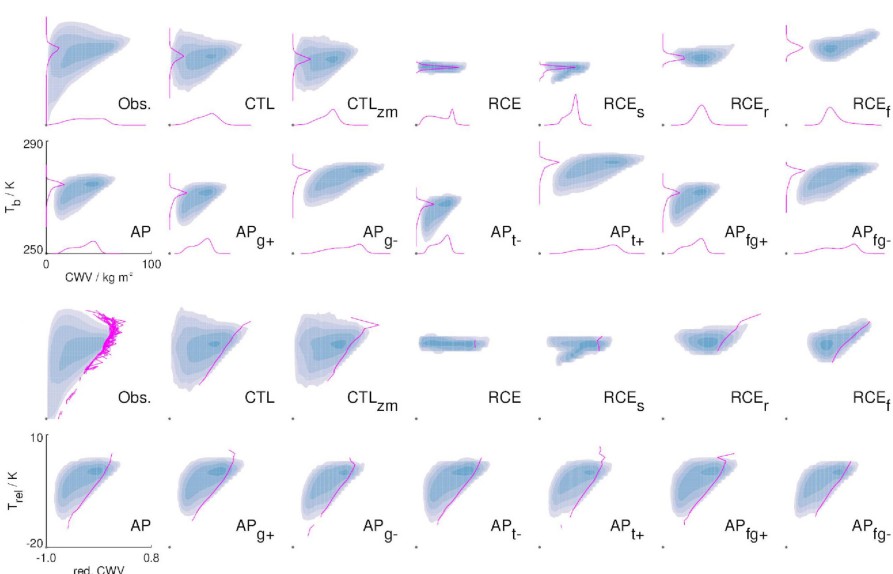

**Figure C1.** The top two rows show the joint probability density distributions of CWV (x-axis) and bulk temperature (y-axis). Pink lines along the x-axis show the probability density distribution of CWV, and pink lines along the y-axis the probability density distribution of bulk temperature. All plots use linear axes with the same axes range and the same normalizations for the pink lines. The blue shading marks probabilities of $10^{-4}, 10^{-3}, 10^{-2}, 10^{-1}, 1$ in %. The bottom two rows are like the top two rows but display the space of reduced CWV (x-axis) and relative temperature (y-axis). Pink lines show the precipitation onset curves. The observations panel contains lines for all years.

 **Appendix D: Cluster properties**

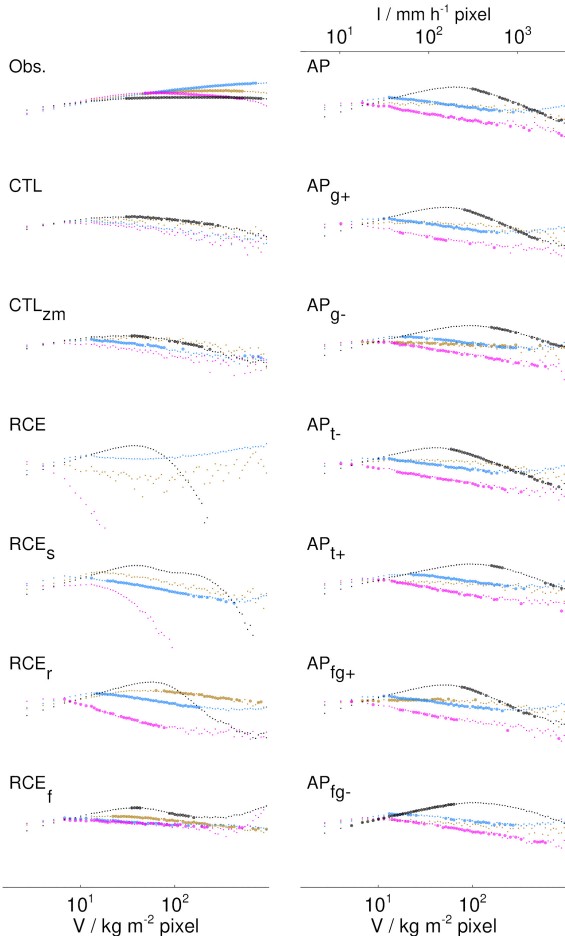

**Figure D1.** Occurrence frequencies of precipitation volume $I$ or CWV volume $V$ belonging to precipitation clusters (black) and CWV islands at the 40% threshold (brown), the 80% threshold (blue), and the critical threshold (magenta). Thick dots mark the intervals with the best power law. All curves are compensated by $(I$ or $V)^{1.69}$. Colors are as in Fig. 7.

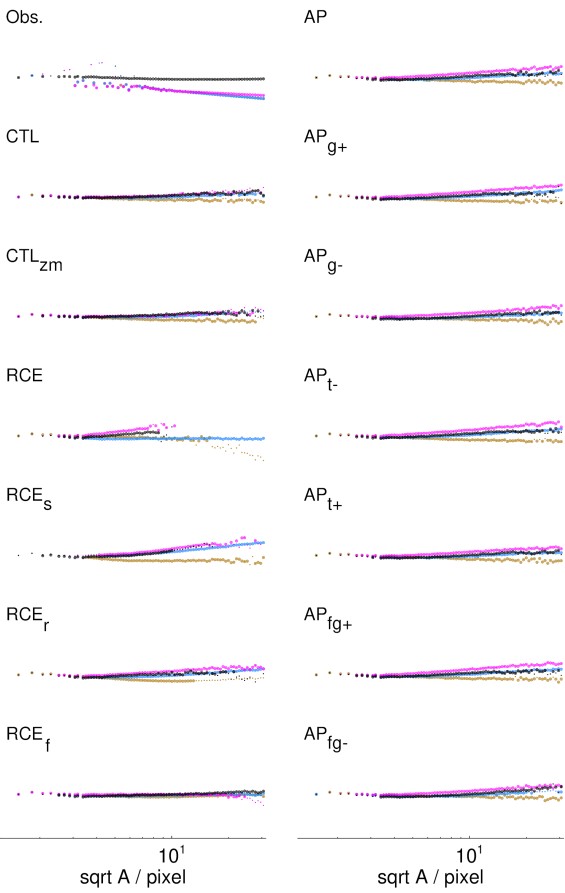

**Figure D2.** Fractal dimensions of precipitation clusters and CWV islands. Shown is circumference $\lambda$ versus $\sqrt{A}$. All spectra are compensated by $\sqrt{A}^{-1.36}$. Colors are as in Fig. 7.

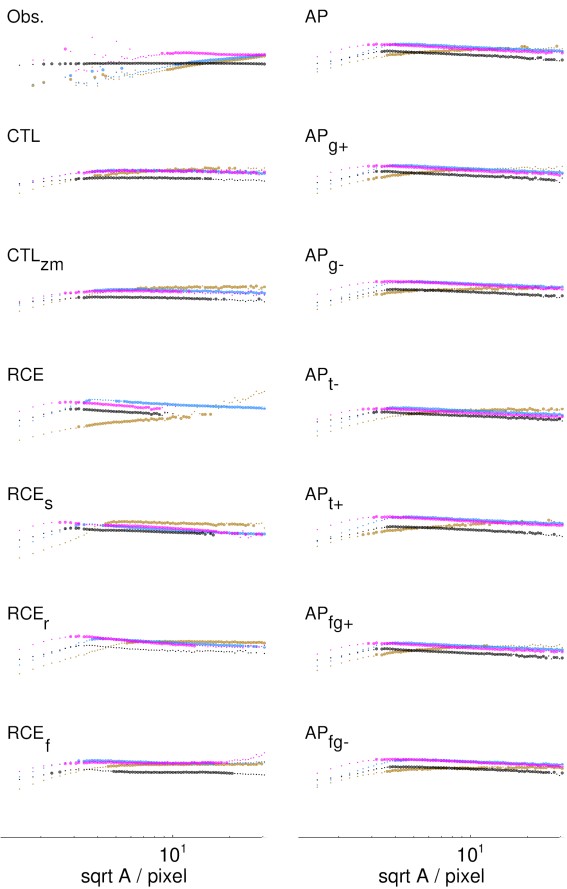

**Figure D3.** Fractal dimensions of precipitation clusters and CWV islands. Shown is precipitation volume $I$ or CWV volume $V$, respectively, versus $\sqrt{A}$. All spectra are compensated by $\sqrt{A}^{-2.26}$. Colors are as in Fig. 7.

*Author contributions.* The authors contributed equally to the conceptualization of the work. CCS performed most of the analysis and writing of the original draft with inputs from BS.

*Competing interests.* The authors declare that they have no competing interests.

*Acknowledgements.* CCS carried out the research when she was still working at the Max Planck Institute for Meteorology and supported by
340 the Minerva Fast Track programme of the Max Planck Society.

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
