# Peer review of "Dynamical imprints on precipitation cluster statistics across a hierarchy of high-resolution simulations"

_EGUsphere, 2024_

## Referee Comment (RC2)

**Dynamical imprints on precipitation cluster statistics across a hierarchy of high-resolution simulations**

Claudia Christine Stephan and Bjorn Stevens

October 1, 2024

**General comments**

This well-written manuscript presents an analysis of high-resolution global simulations to investigate the mechanisms driving the power-law behavior of tropical precipitation clusters. Using a hierarchy of ICON simulations with varying degrees of realism, the authors effectively demonstrate that the presence of stirring and large-scale vertical overturning dynamics, such as planetary and synoptic-scale variability, are key in producing the observed power-law distributions in precipitation statistics.

I believe this study has the potential to shed light on important open questions in the literature regarding the statistical behavior of precipitation clusters and their link to large-scale dynamics. The methodology is sound, the figures are clear, and the article is well-structured.

However, my main concern with the manuscript is the lack of emphasis on the motivation for studying these power-law behaviors and the underlying mechanisms. The authors should better articulate how this work advances the understanding of these phenomena and their significance to the broader scientific community. Additionally, there is a disconnect between the figures and the text. While the figures have the potential to convey a cohesive story, the text provides only a brief discussion, which limits their impact. A more in-depth discussion of the figures would strengthen the manuscript.

**1 Specific comments**

- L59–L63: While the authors state that a 10 km resolution is sufficient to resolve convection, how might the results change with a finer resolution? For example, would a higher resolution capture more small precipitation clusters or column water vapor (CWV) islands, and would the power-law behavior still hold under these conditions?

- L64–L86: Rather than merely describing the technical details of each simulation, the authors should clarify the motivation behind selecting these specific simulations. What scientific questions or hypotheses are addressed by each choice?

- L100: Could the authors provide more context regarding the choice of the 2 mm/hr threshold, especially in relation to thresholds used in other studies?

- L102: Is there any anticipated sensitivity of the results to the definition of pixel connectivity in the analysis? It would be helpful if the authors could discuss this aspect.

- L130: The claim that the CTL simulation closely resembles observations feels somewhat overstated, particularly given notable differences in regions like the Maritime Continent and Australia. It might be more accurate to soften this comparison.

- L142: The statement, "Vertical velocity spectra are useful for comparing the prevalence of different scales of vertical motion between the simulations," is key for understanding a major part of the analysis. The authors should expand on this and explain the significance of different slopes in the vertical velocity spectrum in greater detail.

- L154: Could the authors clarify the motivation for defining the reduced CWV?

- L215: In line with my earlier comment on the study's motivation, the conclusion section could more clearly emphasize the contributions of this work and suggest specific questions or directions for future research that stem from the findings of this study.

**2 Technical corrections**

- L69: typo "input4MIPS"?

- L108: ..."Perimeter $\lambda$ and..." , you introduce $\lambda$ as the perimeter in L104, so it would be clearer to define $\lambda$ there when you first mention it.

- Figure 6: The x-axis label should explicitly say "max reduced CWV" to align with the figure caption.

- Figure 8: Specify which panels correspond to the slopes derived from Figs. 7, C1–C3 to enhance clarity.

---

## Author Comment (AC1)

We thank the reviewers for their constructive comments, which have allowed us to improve the quality of the manuscript. We have addressed the comments and incorporated all valuable suggestions in a revised manuscript. We address each comment separately in the following detailed response. Our responses are in blue color.

Response to Reviewer 1:

This manuscript explores the connection between precipitation clusters and large-scale dynamics using a hierarchy of model configurations within ICON. The use of such a hierarchy is particularly unique and an important contribution to the field. The study finds that power laws of the precipitation cluster characteristics are strongly related to those that govern atmospheric motions. While this work represents an important advance in the field, I strongly feel that – as written – the manuscript glosses over a lot of the details and doesn't describe the results presented in the figures sufficiently. I find that arguments that connect the precipitation clusters to atmospheric motion is weak and that more descriptive explanation is needed before it can be accepted for publication.

We thank the reviewer for pointing out the value of our contribution. We can see that our text was too brief and we acknowledge that we did not well introduce the dynamics aspects, which lead to a disconnection between precipitation clusters and atmospheric motion. We fixed these problems by adding a paragraph on dynamics right at the start of the manuscript:
*"Atmospheric motions span all horizontal scales. Horizontal kinetic energy is associated with geostrophically balanced motions, gravity waves and turbulence, and follows robust power laws when plotted against horizontal wavenumber $\kappa$ (Gage, 1979; VanZandt, 1982; Nastrom and Gage, 1985). The spectrum is shallow at global scales, has a slope of $\kappa-3$ at scales between 10,000–40,000 km, and a slope of $\kappa-5/3$ at shorter scales. The $\kappa-3$ portion of the spectrum is explained by quasi-geostrophic turbulence theory and originates from a downscale enstrophy cascade (Charney, 1971). The mesoscale slope of $\kappa-5/3$ originates from a downscale cascade of wave energy (Cho and Lindborg, 2001; Augier and Lindborg, 2013; Li et al., 2023). Furthermore, in a strongly stratified turbulent flow, one can expect nonlinear interactions between waves and the vortical modes (Muller et al., 1986; Waite and Bartello, 2006; Kitamura and Matsuda, 2010). In contrast to horizontal kinetic energy, the spectrum of vertical kinetic energy is relatively flat (Schumann, 2019), i.e. nearly white. Morfa Avalos and Stephan (2023) showed that the vertical kinetic energy spectrum can be derived from the horizontal kinetic energy spectrum using linear gravity wave theory at large scales and mesoscales, and an incompressible, isotropic scaling of the continuity equation at short scales. While the power laws of atmospheric motions are relatively well understood, this is not the case for moisture fields."*
Further, we added text to point out the link between precipitation clusters and atmospheric motion throughout the text, e.g. in Section 3.2: *"In panel (a) we can clearly see the difference between, for instance, RCE and RCEf , the two simulations that differ most in the slope of vertical kinetic energy. RCE contains either dry regions without precipitation or moist regions with isolated convection. Here, vertical velocity has most energy on short scales. RCEf, on the other hand, forms tropical storms and generates strong vertical motions on large scales."*

And Section 3.3: *"We now turn to question if changes in the atmospheric motion spectrum – in this case diagnosed as the vertical velocity spectrum becoming shallower or steeper – change the statistics of CWV islands and precipitation clusters. This implicitly includes the question whether or not precipitation cluster statistics inherit their properties from the statistics of the CWV distribution, as the study by Li et al. (2022) proposed. From the above discussion of Fig. 7 we see that curves of CWV at different thresholds of precipitation align better in some experiments than in others. This says that precipitation does not inherit its statistics from universal CWV statistics in all experiments. However, in some experiments the 4 colored curves in Fig. 7 have similar shapes, which points to self-affinity of the CWV field. It is straightforward to understand why the curves diverge in some experiments. In RCE, RCEs and RCEr, precipitation occurs in moist patches as isolated, small-scale convection, creating updrafts on small horizontal scales. These motions do not require large-scale overturning but*

*can be compensated by downwelling on similarly small scales. Hence, in RCE, RCEs and RCEr, the small-scale dynamics decouple from the large-scale distribution of the CWV field. Weak and slowly varying overturning only shapes large-scale dry and moist patches. In these experiments there is no self-affinity of the CWV distribution."*

We also discuss each figure in more depth now, as will become clear from our responses to both reviewers' comments.

Some
additional minor comments are below:
L72-73: Is the time-averaged zonal-mean SST just for 1979? Or is it a climatology of a different time period?
We changed the sentence to make it clear: *"CTLzm is like CTL except that it uses constant and zonally averaged SSTs, which correspond to the time-averaged zonal-mean SSTs of CTL, i.e. SSTs of 1979."*

L78-79: I think it is important to point to other studies in the paper that have used similar approaches, such as:
Shi and Bretherton (2014): https://doi.org/10.1002/2014MS000342
Reed and Chavas (2015): https://doi.org/10.1002/2015MS000519
Arnold and Randall (2015): https://doi.org/10.1002/2015MS000498
Chavas and Reed (2019): https://doi.org/10.1175/JAS-D-19-0001.1
We now refer to all of the above in the text.

L78-79: I feel that RCE_r should really be referred to as RCE_f since using constant Coriolis parameter mimics an f-plane.
This is indeed very logical. We changed the label as suggested.

L89: How do you know that 260 days is sufficient for the analysis presented?
We now clarify this by mentioning that the statistics are stable during the 260 days: *"Although the characteristics of the precipitation clusters are stable during these 260 days, we need this amount of data to sample the tails of the distributions."*

L92: L87 above states that the data is output daily for the models, but here it is stated that hourly data is used for the observations. Is that right? How is this rectified?
We added an explanation: *"We use hourly data for the observations, as unlike in the majority of our numerical simulations, there is a diurnal cycle in reality."*

L93: Why 30-min accumulations? If you are comparing to the model output, and the model output is instantaneous, should you try and match the length of the model timestep for the time scale?
We added an explanation: *"Thirty minutes is the shortest available accumulation time for CMORPH. For the model output we chose to analyze instantaneous fields to facilitate the interpretability when connecting different variables, since each variable has a different de-correlation time scale."*

L102: Should it say: "defined as *at least* a four-point connected"
Our formulation was confusing. We changed the text to: *"A cluster is then defined as all pixels that are connected by at least one common edge, i.e. touching corners do not count."*

L105: Since a cluster has a minimum of four connected pixels, you should use a different example than a single pixel here.
This no longer applies.

L141: Wording is confusing/incorrect here: "same modulo differences."

We changed the text to: *"We do not show the extra AP simulations in Fig. 1, because their structures look very similar to AP. Differences between the AP experiments are discussed in Appendix A."*

Figure 4: T_b on y-axis is not defined.
We fixed this.

L173-L179: Is there any connection between the scaling seen here and that typical for Kinetic Energy in the atmopshere? See Nastrom and Gage (1985): https://doi.org/10.1175/1520-0469(1985)042%3C0950:ACOAWS%3E2.0.CO;2 Also, note, there is some similarity between the slopes seen here and the -5/3 slope in Nastrom and Gage (1985) for mesoscale dynamics and turbulence.

No, there is no such link. Why the slope may be what it is remains an open question but we now discuss this in greater depth.

Response to Reviewer 2:

(the text of the pdf could not be copied). Hence, the comments are numbered below the response and we here refer to this numbering.

Response to general comments: We thank the reviewer for pointing out the importance of our contribution. We added text to the introduction (see response to Reviewer 1, "general") and we also clarified our goals in the introduction: *"Even though the modeling study of Ahmed and Neelin (2019), discussed above, could not isolate a clear physical mechanism to explain the cluster scaling laws, it demonstrated that the observed scalings can be obtained in a spatially two-dimensional model, i.e. without explicitly considering the vertical dimension, which would be consistent with spatially two-dimensional KPZ dynamics. Should the KPZ equation be applicable to Earth's atmosphere, then Earth's motion spectrum would have to match the required structure of the noise term η. While a physical interpretation of the η term is not the goal of our study, we want to test if changes in the atmospheric motion spectrum go along with changes in the statistics of precipitation clusters. If this were systematically the case, then it would be strong evidence that the characteristic of precipitation clusters result from the robust scaling laws of atmospheric horizontal (or vertical) kinetic energy. In this study, we therefore consider atmospheres that differ substantially in their dynamic characteristics. The planetary configuration determines the scaling behavior of the atmospheric motion field. Our main result is that self-affinity in the CWV field and scaling laws in precipitation clusters only emerge in atmospheres with large-scale mixing, which suggests that precipitation clusters do indeed inherit their scalings from atmospheric motion spectra."*

We also discuss each figure in more depth now, as will become clear from our responses to both reviewers' comments.

Comment 1: We added text: *"Bravo et al. (2024) tested the convergence of aquaplanet simulations using ICON with the same physics package as our study. They refined the horizontal grid from 160 km down to 1.25 km and report that tropical precipitation and precipitable water converge already at 10 km with only small changes towards finer resolution."*

Comment 2: Done as requested.

Comment 3: We added the following text: *"Models with an explicit representation of moist convection at what is still rather coarse resolution tend to heavy rain (Becker et al., 2021). We thus select 2 mm*

*h−1 closer to the upper bound reported by Li et al. (2022). As we show later, this threshold also produces the expected scalings in observations.”*

Comment 4: We clarified our definition of connectivity: *“A cluster is then defined as all pixels that are connected by at least one common edge, i.e. touching corners do not count.”* Clearly, this is the only logical definition.

Comment 5: We softened the wording to: *“The control simulation CTL with realistic SSTs, land and a diurnal cycle, shows some characteristics that are also seen in observations, such as enhanced moisture above the warm SSTs of the western tropical Pacific.”*

Comment 6: See response to Reviewer 1, “general”: We added a paragraph on dynamics right at the start of the manuscript. Further, we modified the text in Section 3.1: *“Recall that we aim at establishing a link between the statistical properties of the CWV field and those of atmospheric motions. For this reason we compute how vertical kinetic energy is distributed as a function of the zonal wavenumber $\kappa$. A steep slope (large negative values) indicates that motions on large horizontal scales (small $\kappa$) are more energetic than motions on short horizontal scales (large $\kappa$). When the slope is shallow (small negative values), then motions on short scales become relatively more energetic. Hence, vertical velocity spectra are useful for comparing the prevalence of different scales of vertical motion between the simulations.”*

Comment 7: We clarified this by adding text to Section 3.2 and by adding a figure to the appendix (C1): *“Because warmer air can hold more moisture, the CWV threshold for the onset of precipitation depends strongly on tropospheric bulk temperature (Tb), which is defined as mass weighted temperature between 1 and 10 km height. In each simulation, Tb varies in space and time, but due to the different setups of the experiments, the average Tb is also different from experiment to experiment. The most frequently occurring value of Tb, the mode Tm, is shown on the y-axis of Fig. 4. The corresponding x-axis shows the critical CWV, defined as the CWV where precipitation rates reach on average 2 mm h−1 at points with the respective Tm. Since we are not interested in the values of Tb and CWV themselves, but want to focus on the structure of the CWV field and its relationship to dynamics, we rescale Tb and CWV to achieve a better comparability of the simulations. We rescale temperature by subtracting Tm from Tb and we define reduced CWV as (CWV−CWVcrit)/CWVcrit, following Peters and Neelin (2006), who showed that precipitation onset curves collapse when plotted against reduced CWV. This is also true for our simulations. Figure 5 shows the precipitation onset lines in the original phase space (CWV and Tb) and rescaled phase space (reduced CWV and Trel = Tb −Tm). Except for dry or moist outliers, the onset curves collapse onto the same line in the normalized phase space. This is particularly clear for the AP simulations (Fig. 5c,d). For the other simulations (Fig. 5a,b) the change is less impressive because these simulations differ widely in their temperature distributions (the range of simulated temperatures is shown as vertical lines next to the y-axis). Given that CWVcrit scales approximately linearly with Tm (Fig. 4), it may seem surprising that the onset curves collapse only in normalized phase space. An explanation can be found in Fig. C1. For example, from APt− to APt+ the temperature probability density distribution shifts to higher values with little change to its shape. In contrast, the CWV distribution does not merely shift to higher values, but also broadens. This results in a flatter slope of the onset curve. Rescaling the CWV accounts for the width of the CWV distribution.”*

Comment 8: We added text to the discussion: *“Our study shows that precipitation cluster statistics derive their apparent universality from the robust spectral characteristics of atmospheric wave and turbulence dynamics. This is plausible because there has to be a global organization mechanism for precipitation clusters if they obey scaling laws. While this result is new to the best of our knowledge, it is only a qualitative statement and future studies are needed to address the causal chain of underlying mechanisms. Interesting open questions that our study inspires are, for instance:*

*– What is the required slope in vertical velocity to achieve self-similarity in the CWV distribution and what would happen if the slope were steeper?*

*– Is there an important feedback from convection, i.e. is the generation of waves and vertical motions by condensation relevant for the relationship between atmospheric motions and the distribution of CWV?*

*– Why is the roughness exponent close to the prediction of KPZ dynamics? Is it a coincidence or can we interpret the η term of the KPZ equation in a physically meaningful way?*

*We are confident that progress is possible with well-designed numerical experiments. A starting point could be to prescribe atmospheric motions and use water vapor as a passive tracer."*

Comment 9: Clarified.

Comment 10: Done as requested.

Comment 11: Done as requested.

Comment 12: Done as requested.

**Dynamical imprints on precipitation cluster statistics across a hierarchy of high-resolution simulations**

Claudia Christine Stephan and Bjorn Stevens

October 1, 2024

**General comments**

This well-written manuscript presents an analysis of high-resolution global simulations to investigate the mechanisms driving the power-law behavior of tropical precipitation clusters. Using a hierarchy of ICON simulations with varying degrees of realism, the authors effectively demonstrate that the presence of stirring and large-scale vertical overturning dynamics, such as planetary and synoptic-scale variability, are key in producing the observed power-law distributions in precipitation statistics.

I believe this study has the potential to shed light on important open questions in the literature regarding the statistical behavior of precipitation clusters and their link to large-scale dynamics. The methodology is sound, the figures are clear, and the article is well-structured.

However, my main concern with the manuscript is the lack of emphasis on the motivation for studying these power-law behaviors and the underlying mechanisms. The authors should better articulate how this work advances the understanding of these phenomena and their significance to the broader scientific community. Additionally, there is a disconnect between the figures and the text. While the figures have the potential to convey a cohesive story, the text provides only a brief discussion, which limits their impact. A more in-depth discussion of the figures would strengthen the manuscript.

**1 Specific comments**

1 • L59–L63: While the authors state that a 10 km resolution is sufficient to resolve convection, how might the results change with a finer resolution? For example, would a higher resolution capture more small precipitation clusters or column water vapor (CWV) islands, and would the power-law behavior still hold under these conditions?

2 • L64–L86: Rather than merely describing the technical details of each simulation, the authors should clarify the motivation behind selecting these specific simulations. What scientific questions or hypotheses are addressed by each choice?

3 • L100: Could the authors provide more context regarding the choice of the 2 mm/hr threshold, especially in relation to thresholds used in other studies?

4 • L102: Is there any anticipated sensitivity of the results to the definition of pixel connectivity in the analysis? It would be helpful if the authors could discuss this aspect.

5 • L130: The claim that the CTL simulation closely resembles observations feels somewhat overstated, particularly given notable differences in regions like the Maritime Continent and Australia. It might be more accurate to soften this comparison.

6 • L142: The statement, "Vertical velocity spectra are useful for comparing the prevalence of different scales of vertical motion between the simulations," is key for understanding a major part of the analysis. The authors should expand on this and explain the significance of different slopes in the vertical velocity spectrum in greater detail.

7 • L154: Could the authors clarify the motivation for defining the reduced CWV?

8 • L215: In line with my earlier comment on the study's motivation, the conclusion section could more clearly emphasize the contributions of this work and suggest specific questions or directions for future research that stem from the findings of this study.

**2 Technical corrections**

9 • L69: typo "input4MIPS"?

10 • L108: ..."Perimeter $\lambda$ and..." , you introduce $\lambda$ as the perimeter in L104, so it would be clearer to define $\lambda$ there when you first mention it.

11 • Figure 6: The x-axis label should explicitly say "max reduced CWV" to align with the figure caption.

12 • Figure 8: Specify which panels correspond to the slopes derived from Figs. 7, C1–C3 to enhance clarity.